# The Chemopreventive Effects of Polyphenols and Coffee, Based upon a DMBA Mouse Model with microRNA and mTOR Gene Expression Biomarkers

**DOI:** 10.3390/cells11081300

**Published:** 2022-04-12

**Authors:** Richard Molnar, Laszlo Szabo, Andras Tomesz, Arpad Deutsch, Richard Darago, Bence L. Raposa, Nowrasteh Ghodratollah, Timea Varjas, Balazs Nemeth, Zsuzsanna Orsos, Eva Pozsgai, Jozsef L. Szentpeteri, Ferenc Budan, Istvan Kiss

**Affiliations:** 1Doctoral School of Health Sciences, Faculty of Health Sciences, University of Pécs, 7624 Pécs, Hungary; laszlo.szabo.pte@gmail.com (L.S.); andras.tomesz.pte@gmail.com (A.T.); deutscharpad@gmail.com (A.D.); daragorichard@gmail.com (R.D.); raposa.bence@gmail.com (B.L.R.); 2Department of Public Health Medicine, Medical School, University of Pécs, 7624 Pécs, Hungary; taytakh@yahoo.com (N.G.); vtimi_68@yahoo.com (T.V.); nem_bal2@hotmail.com (B.N.); zsuzsa.orsos@aok.pte.hu (Z.O.); pozsgay83@gmail.com (E.P.); istvan.kiss@aok.pte.hu (I.K.); 3Institute of Transdisciplinary Discoveries, Medical School, University of Pécs, 7624 Pécs, Hungary; 4Institute of Physiology, Medical School, University of Pécs, 7624 Pécs, Hungary

**Keywords:** miR, carcinogen, 7,12-dimethylbenz[a]anthracene, polyphenol, coffee, miR-134, miR-132, miR-124-1, miR-9-3, mTOR

## Abstract

Polyphenols are capable of decreasing cancer risk. We examined the chemopreventive effects of a green tea (*Camellia sinensis*) extract, polyphenol extract (a mixture of blackberry (*Rubus fruticosus*), blackcurrants (*Ribes nigrum*), and added resveratrol phytoalexin), Chinese bayberry (*Myrica rubra*) extract, and a coffee (*Coffea arabica*) extract on 7,12-dimethylbenz[a]anthracene (DMBA) carcinogen-increased miR-134, miR-132, miR-124-1, miR-9-3, and *mTOR* gene expressions in the liver, spleen, and kidneys of CBA/Ca mice. The elevation was quenched significantly in the organs, except for miR-132 in the liver of the Chinese bayberry extract-consuming group, and miR-132 in the kidneys of the polyphenol-fed group. In the coffee extract-consuming group, only miR-9-3 and mTOR decreased significantly in the liver; also, miR-134 decreased significantly in the spleen, and, additionally, miR-124-1 decreased significantly in the kidney. Our results are supported by literature data, particularly the DMBA generated ROS-induced inflammatory and proliferative signal transducers, such as TNF, IL1, IL6, and NF-κB; as well as oncogenes, namely *RAS* and *MYC*. The examined chemopreventive agents, besides the obvious antioxidant and anti-inflammatory effects, mainly blocked the mentioned DMBA-activated factors and the mitogen-activated protein kinase (MAPK) as well, and, at the same time, induced *PTEN* as well as *SIRT* tumor suppressor genes.

## 1. Introduction

Nowadays, the incidence and mortality of cancer in high-income countries (HIC) is decreasing [1,2], but in the low- and middle-income countries (LMIC), the trend-line is still supposed to increase slightly [1]. According to the WHO’s assessment, 30–50% of cancer cases could have been prevented [3]. Indeed, the improving tendency in HIC is the result of successful primer prevention, early detection, and advanced therapies [1]. However, cancer is still, globally, the second leading cause of death (with approximately 9.6 million deaths in 2018) [4] and is also the greatest disease burden.

Therefore, novel chemopreventive strategies are warranted to enhance anticarcinogen mechanisms [5,6,7,8,9,10]. In vitro studies and in vivo animal experiments suggest antimutagenic and anticarcinogenic effects of flavonoids [11,12,13]. Moreover, flavonoids are consumed widely, and a negative correlation was found between the total flavonoid intake and the incidence of lung cancer formation among smokers [11].

Among flavonoids, green tea catechin (GTC), polyphenols [14], myricetin (3,5,7,3′,4′,5′-hexahydroxyflavone) [15], and stilbene resveratrol (3,5,4′-trihydroxystilbene), as well as its precursor, the piceid (3,5,4′-trihydroxystilbene-3-O-‚-D-glucopyranoside) [12], are promising compounds. The anticancer properties of these compounds have been proven by numerous in vitro and in vivo experiments, as well as clinical and epidemiological studies [9,13,16]. In addition, the main components of coffee (*Coffea arabica*), such as caffeine, chlorogenic acids, hydroxycinnamic acids, melanoidins, etc., also exert tumor-suppressive effects [16,17]. However, during the Maillard reactions when coffee is roasted (besides the melanoidins), acrylamide and furan are produced in traces [17,18], which are second-class carcinogens [17].

Grapes (*Vitis vinifera*) are abundant in stilbene phytoalexin molecules, such as resveratrol and piceid, among other polyphenols, namely anthocyanins, flavanols, flavonols, etc. [12,19]. Although resveratrol’s bioavailability is poor, it provided promising anticarcinogen results in preclinical in vitro and in vivo animal tests [20]. Still in a clinical phase I pilot study, the cancer-preventative effects of resveratrol and freeze-dried grape powder were confirmed, as they significantly inhibited (*n* = 8, *p* < 0.03) the expression of the WNT oncogene in the colonic mucosa [21]. Catechins of green tea (*Camellia sinensis*) are among the main anticarcinogenic chemopreventive agents [14], especially the most potent epigallocatechin-3-gallate (EGCG) [14,22,23]. A meta-analysis highlighted that daily consumption of green tea decreased the risk of liver cancers [22] in Asian women (with 5+ cups consumed daily) in a significant manner [23]. Chinese bayberry (*Myrica rubra*) contains a high amount of myricetin (3,5,7,3′,4′,5′-hexahydroxyflavone), that can be extracted [24]. Based on nutrition surveys, the myricetin intake decreased the relative risk (RR) of prostate cancer between the highest and lowest quartiles of myricetin-consuming men (RR = 0.43 (95% CI: 0.22, 0.86: *p* for trend = 0.002)) [15]. According to the case-controlled study by Ferruci et al., the regularly high intake of tea combined with coffee reduced the risk of basal cell carcinoma (BCC), compared to proper random control persons (OR = 0.57, 95% CI = 0.34–0.95, *p* = 0.037) [25].

Further in vivo tests are required, with the purpose of possessing knowledge about the chemopreventive agent’s possible additional health-promoting effects in order to develop novel chemopreventive strategies. To perform an in vivo test, the 7,12-dimethylbenz[a]anthracene (DMBA) carcinogen model was utilized [26]. DMBA is a complete, pluripotent carcinogen aromatic hydrocarbon molecule [27], which forms DNA adducts and generates reactive oxygen species (ROS) [28,29], inducing carcinogenesis [30]. Therefore, DMBA causes, in codon 61 of the RAS oncogenes CAA→CTA, transversion mutations [31]. Moreover, DMBA alters the expression patterns of onco- and tumorsuppressor genes in the following manner: DMBA increases the expression levels of oncogenes, which consequently (in most cases) increases the expression of protective tumorsuppressor genes [26,32,33,34,35]. Moreover, in vivo experimental data has established some relevant miRNAs (miRs), such as miR-134, miR-132, miR-124-1, and miR-9-3, whose expressions are increased in response to DMBA exposure [36,37,38,39,40]. MiRs are noncoding, single-stranded RNA molecules transcribed from DNA. After maturation, their length is 19–25 nucleotides and they are transported to target cells by the following carriers: apoptosis bodies, exosomes, membrane-derived vesicles, high-density lipoproteins (HDL), and ribonucleoprotein complexes [41]. MiRNAs bind to their target mRNA complementary sequences in the 3′-untranslated region (3′-UTR) of a protein-coding gene, leading to a decrease in protein synthesis [42]. In chemical carcinogen-induced tumorigenesis, dysregulated patterns of miRNAs play crucial roles [41]. For example, DMBA exposure increases the expression of the oncogene *RAS* family [35], while miRs play an important role in (RAS-involved) mitogen-activated protein kinase (MAPK) pathways, inducing carcinogenesis [43].

Thus, the mentioned molecular epidemiological biomarkers indicate DMBA exposure early and in a reliable manner [26,32,33,35,39,40]. Furthermore, we also examined the gene expression level of the *mammalian target of rapamycin* (*mTOR*), which is involved in several cellular homeostasis mechanisms [36,38]. More specifically, the liver, kidneys, and spleen parenchymal organs were studied because the DMBA treatment caused a significantly increased expression on relevant *miR-134*, *miR-132*, *miR-124-1*, *miR-9-3*, and *mTOR* in those organs for at least 24 h, based upon earlier research data [36,37,38]. Moreover, in earlier studies, the examined chemopreventive polyphenol extract, green tea extract, Chinese bayberry extract, and coffee extract ameliorated the DMBA, caused repetitive long interspersed element-1 (LINE-1) DNA hypomethylation [9].

In this study, the preventive effects of several polyphenols, namely the green tea extract (catechin content of 80%), Chinese bayberry extract (myricetin content of 80%), polyphenol extract (with 4 g/100 mL added to resveratrol), and coffee extract were examined in a DMBA-treated mouse model to elucidate the effects of chemopreventive agents on the expression profile of the mentioned *miRs* and *mTOR*, in order to decide if their elevation caused by DMBA exposure can be mitigated or not.

## 2. Materials and Methods

### 2.1. Animal Treatment

The experimental setting in our study was similar to that described by Szabo et al. 2021 [9]. We utilized six groups of female CBA/Ca mice (*n* = 6) aged 12 weeks. Pre-feeding was not given to the untreated and DMBA-treated control groups; however, one group received 4 mg/day of the animal green tea (*Camellia sinensis*) excerpt (Xi’an Longze Biotechnology Co. Ltd., Xi’an, China); one group received 2.5 mg/day of the animal Chinese bayberry (Myrica Rubra) supplement (Xi’an Longze Biotechnology Co. Ltd., Xi’an, China); one group received 30 mg/day of the animal polyphenol extract (common grapevine (*Vitis vinifera* ‘Cabernet Sauvignon’) seed and peel, blackberry ‘thorn free’ (*Rubus fruticosus* ‘Thornfree’) seed and peel, and blackcurrants, plus an additional 4 g/100 mL of resveratrol, in particular, FruitCafe^TM^ (Slimbios Ltd., Budapest, Hungary); and one group received a coffee (*Coffea arabica*) extract for two weeks (30 mg/day/animal, up to 150 mL) in addition to their regular feed. All other five classes of animals received 20 mg/bwkg DMBA intraperitoneally (Sigma-Aldrich, St. Louis, MO, USA), dissolved in 0.1 mL of corn oil, with the exception of the untreated control group. Animals were put to death by cervical dislocation after 24 h of DMBA exposure, and their kidneys, liver, and spleen were extracted. Table 1 summarizes the specifics of the experimental setup, as well as the substances used.

Tomesz et al.’s 2020 publication [36] utilized the same experimental procedure. Animal experimentation standards and criteria were followed when housing mice. All the measures have been taken to avoid unnecessary pain. The experiment was carried out in accordance with current ethical rules (the Animal Welfare Committee of the University of Pécs issued the ethical license no. BA02/2000-79/2017).

### 2.2. Collective Isolation of RNA

A TRIZOL reagent (Thermo Fisher Scientific, Waltham, MA, USA) was used to isolate total cellular RNA, according to the manufacturer’s guidelines. The quality of the RNA was determined using NanoDrop absorption photometry, and only RNA fractions with A > 2.0 at 260/280 nm were utilized in the RT-PCR (reverse transcription polymerase chain reaction) procedure.

### 2.3. Polymerase Chain Reaction in Reverse Transcription (RT-PCR)

On a LightCycler 480 qPCR system (Roche Diagnostics, Indianapolis, IN, USA), one-step PCR, containing a reverse transcription and a target amplification, was done in a 96-well plate using Kapa SYBR FAST One-step qPCR equipment (Kapa Biosystems, Wilmington, MA, USA).

The following temperatures of the program were used:

After a 5-min incubation at 42 °C, a 3-min incubation at 95 °C, 45 cycles (95 °C for 5 s, 56 °C for 15 s, and 72 °C for 5 s) was executed, with a fluorescence reading taken at the finish of each cycle. To improve the specificity of the amplification, a melting curve analysis was done on each run (95 °C for 5 s, 65 °C for 60 s, and 97 °C). The following components were used in the reaction mixture: 10 μL of KAPA SYBR FASTqPCR Master Mix, 0.4 μL of KAPA RT Mix, 0.4 of dUTP, 0.4 μL of primers, and 5 μL of a miR template in a total amount of 20 μL of sterile double-distilled water.

Table 2 summarizes the primer sequences (5′-3′) of the mTORC1 gene, the studied miRs (miR-134, miR-132, miR-124-1, and miR-9-3), and the internal control (the mouse U6 gene). Integrated DNA Technologies (Integrated DNA Technologies Inc., Coralville, Iowa, USA) synthesized the primers, and the sequences were obtained from earlier publications [44,45].

### 2.4. Calculations and Statistical Analyses

The 2^−ΔΔCT^ approach was used to determine and compare relative miR expression levels. The Kolmogorov–Smirnov test, Levene’s test, and the T-probe were used to compare averages and test distributions and variances throughout the statistical study. For computations and analyses, the IBM SPSS 21 statistical program (International Business Machines Corporation, Armonk, NY, USA) was utilized. The statistical standard of significance was set at *p* < 0.05.

## 3. Results

### 3.1. Effect of Flavonoid Extract and DMBA Treatment in the Liver, Spleen, and Kidneys, Compared to the DMBA Positive Control

In the livers of animals, the consumption of the polyphenol extract significantly reduced the expressions of miR-9-3 (−41%; *p* < 0.05; SD = 11.1%), miR-124-1 (−68%; *p* < 0.001; SD = 10.1%), miR-132 (−62.9%; *p* < 0.001; SD = 9.2%), miR-134 (−77.9%; *p* < 0.001; SD = 5.6%), and mTORC1 (−49%; *p* < 0.001; SD = 8.4%) when compared to the positive DMBA control group (Figure 1A). We also observed a significant decrease in the expression of miR-9-3 (−38%; *p* < 0.05; SD = 12.1%), miR-124-1 (−59%; *p* < 0.05; SD = 9.8%), miR-132 (−62.4%; *p* < 0.001; SD = 8%), miR-134 (−60.4%; *p* < 0.001; SD = 8%), and mTORC1 (−39%; *p* < 0.001; SD = 8.6%) in the spleens of animals, compared to the positive DMBA control (Figure 1B). In the kidneys of animals, miR-9-3 (−59%; *p* < 0.001; SD = 7.8%), miR-124-1 (−62%; *p* < 0.05; SD = 13.1%), miR-134 (−81.4%; *p* < 0.001; SD = 3.7%), and mTORC1 (−59%; *p* < 0.001; SD = 6.3%) expressions were significantly lower in response to the polyphenol extract, compared to the positive DMBA control (Figure 1C), while the values for miR-132 (−27.1%; *p* = 0.051; SD = 13.7%) were not statistically significant.

### 3.2. Effect of Green Tea Extract and DMBA Treatment in the Liver, Spleen, and Kidneys, Compared to the DMBA Positive Control

The consumption of the green tea extract significantly reduced the expression of miR-9-3 (−33%; *p* < 0.05; SD = 12.9%), miR-124-1 (−69%; *p* < 0.001; SD = 7.4%), miR-132 (−45.4%; *p* < 0.05; SD = 10.2%), miR-134 (−59.2%; *p* < 0.001; SD = 8.9%), and mTORC1 (−57%; *p* < 0.001; SD = 6.7%) in the livers of animals, compared to the positive DMBA control (Figure 2A). The green tea extract resulted in a decrease in the expression of miR-9-3 (−56%; *p* < 0.001; SD = 8.5%), miR-124-1 (−62%; *p* < 0.001; SD = 11.3%), miR-132 (−61.1%; *p* < 0.001; SD = 9.1%), miR-134 (−47.6%; *p* < 0.05; SD = 11.2%), and mTORC1 (−58%; *p* < 0.001; SD = 5.1%) in the spleens, compared to the positive DMBA control (Figure 2B). In the kidneys, we also observed a significant decrease in the expression of miR-9-3 (−48%; *p* < 0.05; SD = 11.4%), miR-124-1 (−36%; *p* < 0.05; SD = 16.6%), miR-132 (−59.6%; *p* < 0.001; SD = 10.8%), miR-134 (−53.3%; *p* < 0.001; SD = 11.1%), and mTORC1 (−57%; *p* < 0.001; SD = 5.6%) in the group consuming the green tea extract, compared to the positive DMBA control (Figure 2C).

### 3.3. Effect of Chinese Bayberry Extract and DMBA Treatment in the Liver, Spleen, and Kidneys, Compared to the DMBA Positive Control

In the liver, a statistically significant decrease was observed in miR-9-3 (−58%; *p* < 0.001; SD = 9.1%), miR-124-1 (−43%; *p* < 0.05; SD = 14.6%), miR-134 (−40.6%; *p* < 0.05; SD = 16.8%), and mTORC1 (−39%; *p* < 0.001; SD = 9.6%) in the Chinese bayberry extract group compared to the positive DMBA control, while the decrease in miR-132 (−19.1%; *p* = 0.14; SD = 14.9%) was not statistically significant (Figure 3A). There were statistically significant downward changes for miR-9-3 (−46%; *p* < 0.05; SD = 11.1%), miR-124-1 (−57%; *p* < 0.05; SD = 12.9%), miR-132 (−32.3%; *p* < 0.05; SD = 15.1%), miR-134 (−51.8%; *p* < 0.001; SD = 10.3%), and mTORC1 (−32%; *p* < 0.001; SD = 8.6%) in the spleens of the Chinese bayberry extract-consuming group, compared to the positive DMBA control (Figure 3B). In the kidneys, compared to the positive DMBA control, a statistically significant decrease could be observed for miR-9-3 (−40%; *p* < 0.05; SD = 13.2%), miR-124-1 (−51%; *p* < 0.05; SD = 14%), miR-132 (−57.9%; *p* < 0.001; SD = 10.5%), miR-134 (−28.8%; *p* < 0.05; SD = 12.8%), and mTORC1 (−22%; *p* < 0.05; SD = 11.9%) in the Chinese bayberry extract group (Figure 3C).

### 3.4. Effect of Coffee Extract and DMBA Treatment in the Liver, Spleen, and Kidneys, Compared to the DMBA Positive Control

In the livers, we observed a significant decrease in miR-9-3 (−37%; *p* < 0.05; SD = 19.8%) and mTORC1 (−37%; *p* < 0.05; SD = 14%) expressions in the group consuming the coffee extract, compared to the positive DMBA control, while the results for miR-124-1 (−21%; *p* = 0.21; SD = 23.6%), miR-132 (−16.7%; *p* = 0.24; SD = 19.4%), and miR-134 (−12.7%; *p* = 0.32; SD = 16.7%) were not statistically significant (Figure 4A). In the spleens, the expression of miR-9-3 (−46%; *p* < 0.05; SD = 10.7%), miR-134 (−38.9%; *p* < 0.05; SD = 12.7%), and mTORC1 (−20%; *p* < 0.05; SD = 8.9%) showed a statistically significant decrease in the coffee extract-consuming group, compared to the positive DMBA control, while the decrease in the expression of miR-124-1 (−15%; *p* = 0.37; SD = 22.9%) and the slight increase in the expression of miR-132 (13.1%; *p* = 0.40; SD = 23%) was not statistically significant (Figure 4B). In the kidneys, statistically significant decreases could be observed in miR-9-3 (−31%; *p* < 0.05; SD = 12.8%), miR-124-1 (−47%; *p* < 0.05; SD = 13.6%), miR-134 (−31.6%; *p* < 0.05; SD = 13.5%), and mTORC1 (−22%; *p* < 0.05; SD = 8.7%) in the coffee extract group, compared to the positive DMBA control, while the slight increase in miR-132 (22.1%; *p* = 0.18; SD = 25.4%) was not statistically significant (Figure 4C).

Table 3 shows the summary of expression changes caused by feeding in the observed DMBA pretreated organs.

## 4. Discussion

DMBA induces cellular damage by releasing reactive oxygen species (ROS), which triggers the production of cytokines (such as TNF, IL1, IL6) and transcription factors (such as NF-kB) [29,38,46], as well as lowering the protective glutathione (GSH) level [29,38,46,47]. These consequences result in redundantly activated inflammatory and proliferative secondary signal transduction pathways that are self-induced.

According to in vitro studies, resveratrol, EGCG, and myricetin inhibit CYP 1A1 and 1A2 enzymes [48,49,50], which activate DMBA [35]. If the DMBA activation is hindered, then the consequent *HA-RAS* and *C-MYC* oncogene overexpression is also blocked [32].

Polyphenol structures are generally ROS-scavenging antioxidants that also exert anti-inflammatory effects [51,52]. Thus, molecular features of flavonoids [53], chlorogenic acids, hydroxycinnamic acids, and the caffeine content of coffee [54] and melanoidins [17] exert antioxidant (ROS-quenching) effects, directly mitigating the ROS-induced cellular damage [55,56,57,58]. Moreover, resveratrol [59], myricetin [53,58], GTC [60], and chlorogenic acid [61] induce the protective superoxide-dismutase (SOD) and glutathione-S-transferase (GST) enzymes. Furthermore, both resveratrol, as well as chlorogenic acids, decrease IL-1β, IL-6, and TNF-α expressions [62,63] among others. Flavonoids thereby regulate the carcinogen and/or inflammatory effect-activated signal transduction pathways; for example, they inhibit protein tyrosine and focal adhesion kinases, as well as matrix metalloproteinases (MMPs) [57,64].

Resveratrol [62] and myricetin [65] up-regulate cAMP-response element-binding proteins (CREB) through the activated silent Information Regulator T1 (SIRT1)-dependent pathway [66,67] resulting in a decrease in *miR-134*, *miR-124*, and *mTOR* expressions [68]. In contrast, the EGCG inhibited SIRT1, and both EGCG and resveratrol inhibited NF-κB activity as well [62,63,69], while NF-κB generally decreases miR-124 [70] and miR-132 [71]. Theoretically, decreased NF-κB1 activity (in a seemingly mutually exclusive mechanism) increases the expression of the anticarcinogen *miR-134* [72]. However, NF-κB, TNF-α, and IL-1β increase *miR-9* expression, which downregulates *NF-κB* expression in a negative feedback loop [73].

Moreover, in the coffee consuming group, the chlorogenic acids exert a kidney protective effect by inducing miR-134 [74], which suppresses MMP-9 and MMP-7 [75], ultimately decreasing cyclin D1 [76], which is encoded by *CCND1* and is in inverse correlation with miR-134 [74]. In addition, resveratrol [77], EGCG [78], and caffeine [79] induce *phosphatase and tensin homolog (PTEN)* gene activity, which decreases cyclin D1 cell cycle proteins [80] as well. Still, the stronger negative feedback regulation of miRs [36,37,81,82,83] prevail, ultimately decreasing *miR-134* expression [36,38,81].

Resveratrol [13,63] and EGCG [84] inhibit antiapoptotic cascades by suppressing MAPK pathways [84]. This induces cell cycle arrest in the G0/G1 phase, which downregulates miR-132 and upregulates miR-9 [84]. In all examined groups, the presumably decreased cyclin D1 level led to the decrease in *mTOR* expression [85]. In addition, myricetin blocks phosphoinositide 3-kinase (PI3K) [86], ultimately decreasing *mTOR* expression [87].

MiR-132 inhibits the RAS p21 protein activator GTPase activating protein 1 (RASA1), which inhibits *NRAS* expression and *HRAS* activation [88]; thus, miR-132 ultimately supports the MAPK cascade in this context. Therefore, the silencing of *miR-132* expression could mediate the chemopreventive effect of resveratrol and EGCG. However, miR-9 blocks neurofibromin, which inhibits NRAS activation [88]. Thus, the induction of miR-9 seems to contradict the chemopreventive effect of resveratrol and EGCG in this context. The same is true for the pro-inflammatory cytokines’ increased intercellular adhesion molecule-1 (ICAM1) expression [83], which is blocked by resveratrol [89]. Despite that, ICAM1 positively modulates anti-inflammatory *miR-124* expression [83], which inhibits MAPK signal transduction [88]. The MAPK signaling pathway upregulates *C-MYC*, which activates AKT and cyclinD1 [90,91]. Still, CREB downregulates miR-9 strongly in a negative feedback minicircuitry [92], while miR-9 is also negatively correlated with NF-κB1 activity [93], which decreases miR-124 [69], corresponding to the results of this study.

Myricetin in liver cells as a pro-oxidant increases hydroxyl radicals (˙OH) if catalase (CAT) and SOD enzymes are blocked [94]. Aromatic hydrocarbons (such as DMBA) produce singlet molecular oxygen [95] that reacts with the histidine group of CAT and SOD, deteriorating these enzymes [96], leading to further increased ˙OH levels, which induces protective miR-132 expression [97] in coherence with the results of this study.

Moreover, in the coffee consuming group, the traces of acrylamide and furan exert antagonistic effects against the examined chemopreventive agents; namely, acrylamide (≤100 μmol/L) in vitro, which significantly increases the proliferation of human HCC HepG2 cells and induces the EGFR/PI3K/AKT/cyclin D1 pathway, leading to decreased PTEN levels [98]. The epigenetic carcinogen furan also alters relevant cell cycles, as well as the apoptosis regulator gene expression in the rat’s liver [99], and forms metabolites, which decreases GSH levels with chemical reactions [100].

The above-mentioned experimental materials and their decay products, substrates, enzymes, proteins, and signal transducer molecules orchestrate the observed expression patterns of *miRs* and *mTOR*s, as well as influencing the cell proliferation (Figure 5).

## 5. Conclusions

In all the examined organs in the green tea, myricetin, and flavonoid extract-treated groups, the DMBA elevated expression levels of *miR-134*, *miR-132*, *miR-124-1*, *miR-9-3*, and *mTOR* decreased significantly—except for *miR-132* in the liver of the Chinese bayberry extract-consuming group, and *miR-132* in the kidneys of the flavonoid fed group. However, in the coffee consuming group, only *miR-9-3* and *mTOR* decreased significantly in the liver, *miR-134* decreased in the spleen, and additionally, *miR-124-1* decreased in the kidneys (Table 3).

These experimental agents possess similar chemopreventive molecular mechanisms, including ROS scavenging, as well as signal transduction modulating effects; namely, both DMBA induced inflammatory and proliferative pathways were inhibited, presumably through deactivating TNF, IL1, IL6, and NF-κB [29,38,46,101]. According to the literature, chemopreventive agents presumably decrease all expressions of the examined *miRs* and *mTOR* [68,70,71,93] by induced CREB and decreased NF-κB activities [62,63,65,69]. However, miR-134 was expected to increase with decreased NF-κB activity [72,74] and anti-inflammatory mir-124 should have been positively modulated by ICAM1 [83], contradicting our results.

Individual molecular features were indicated also; for example, the liver-specific pro-oxidant effect of myricetin increased only in the liverthe ROS sensitive *miR-132* expression, in comparison with other studied organs [97]. In the coffee consuming group, the effects of beneficial flavonoids, chlorogenic acids, and melanoidins [17] were most likely partly antagonized by the carcinogen acrylamide and furan content of coffee [98,99].

Moreover, the results could be deceptive, since in the late stages, malignant tumors mostly also downregulated anticarcinogen miRs, for example, miR-134 in invasive and metastatic HCC and RCC [102], or both miR-124 and miR-134 in glioblastoma, and miR-124 in squamous cell carcinoma [88]. However, miR-9 is upregulated in glioma [88]. Therefore, we can suppose that expression levels of *mTOR* and *miRs* are biomarkers, rather than relevant signal transductors, in this context [103].

In summary, the novel finding of this study is that the expression patterns of miR-9-3, miR-124-1, miR-132, miR-134, and *mTOR*, as molecular epidemiological biomarkers, indicated the early carcinogen effect of DMBA and the anticarcinogen effects of the polyphenol extract, green tea extract, Chinese bayberry extract, and coffee extract, which are chemopreventive agents against DMBA exposure, in accordance with the specific molecular features of the contained compounds. Our results contribute to the research of chemoprevention by assuming that the regular consumption of a diet abundant in polyphenols, as well as coffee, exerts anti-inflammatory and anti-cancer effects. These assumptions may form further investigations to improve our eating habits.

## Figures and Tables

**Figure 1 cells-11-01300-f001:**
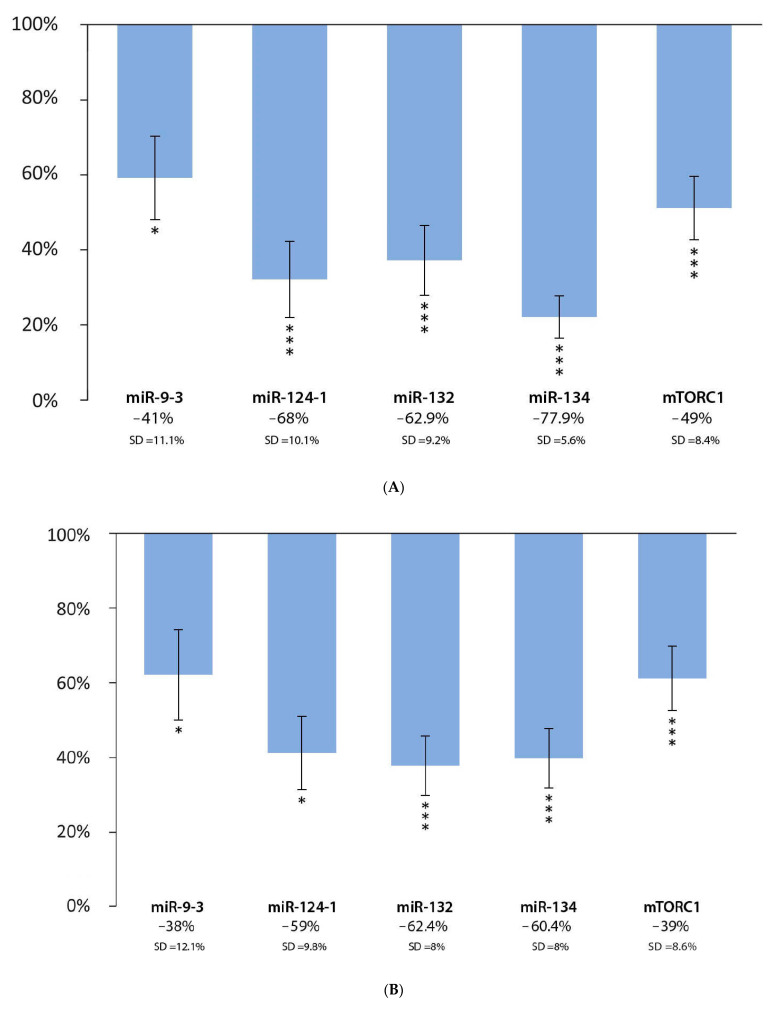
Expression patterns of miR-9-3, miR-124-1, miR-132, miR-134, and mTORC1 in the liver (**A**), spleen (**B**), and kidneys (**C**) of mice treated with DMBA and polyphenol extract (*n* = 6), compared to the DMBA-induced (*n* = 6) expression (* *p* < 0.05; *** *p* < 0.001).

**Figure 2 cells-11-01300-f002:**
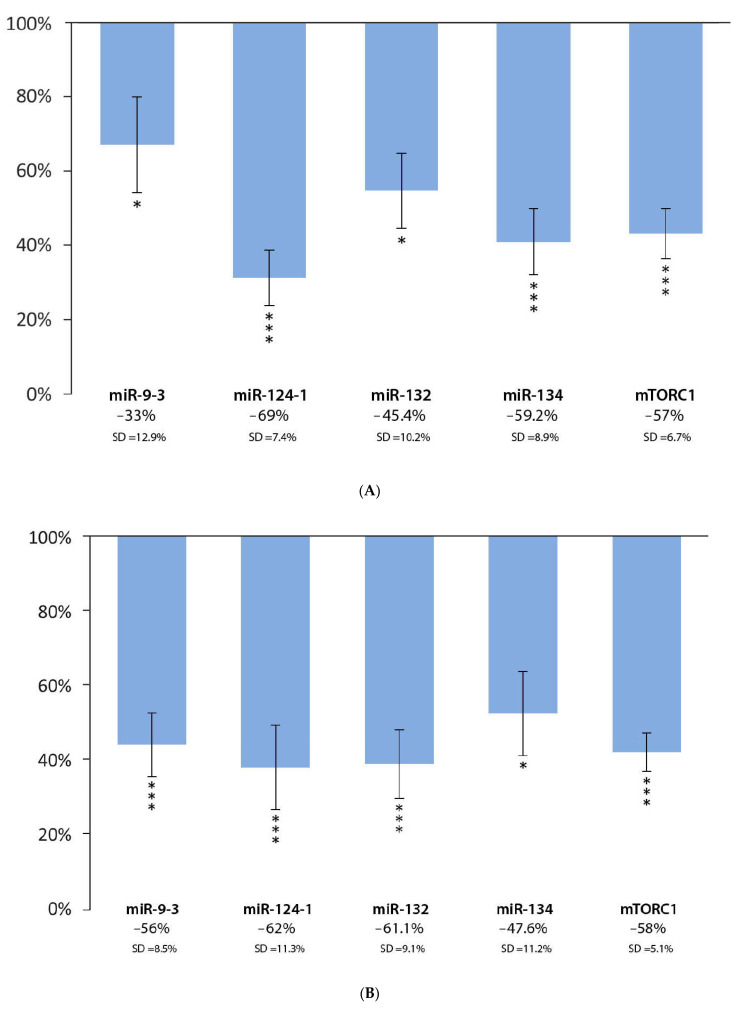
Expression patterns of miR-9-3, miR-124-1, miR-132, miR-134, and mTORC1 in the liver (**A**), spleen (**B**), and kidneys (**C**) of mice treated with DMBA and green tea extract (*n* = 6), compared to the DMBA-induced (*n* = 6) expression (* *p* < 0.05; *** *p* < 0.001).

**Figure 3 cells-11-01300-f003:**
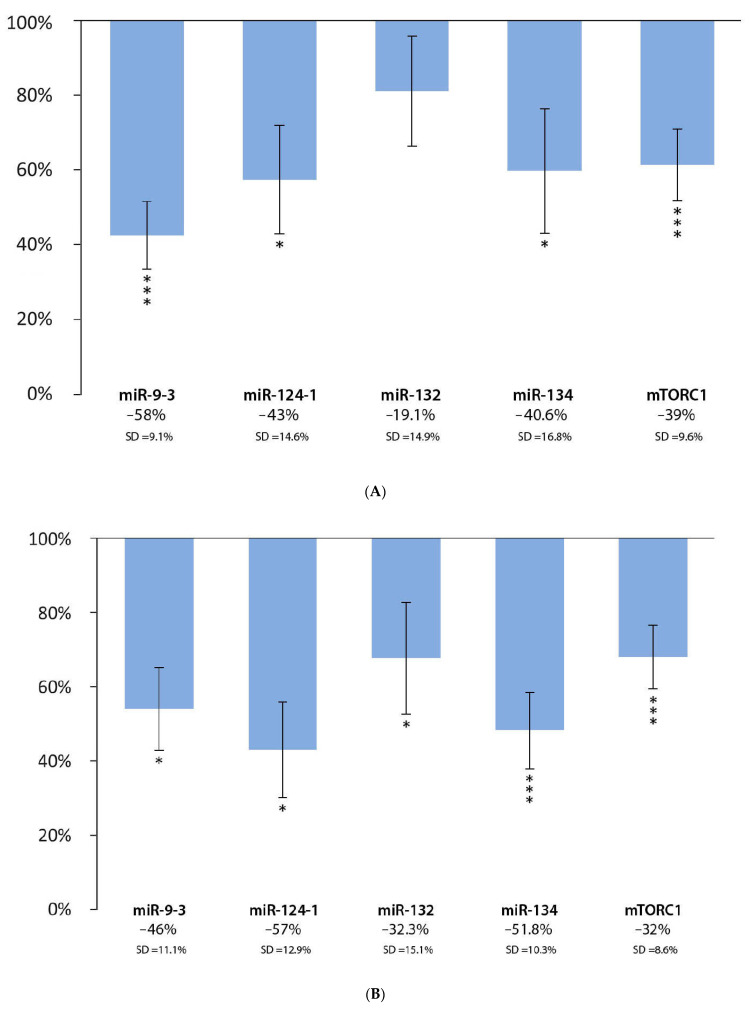
Expression patterns of miR-9-3, miR-124-1, miR-132, miR-134, and mTORC1 in the liver (**A**), spleen (**B**), and kidneys (**C**) of mice treated with DMBA and Chinese bayberry extract (*n* = 6), compared to the DMBA-induced (*n* = 6) expression (* *p* < 0.05; *** *p* < 0.001).

**Figure 4 cells-11-01300-f004:**
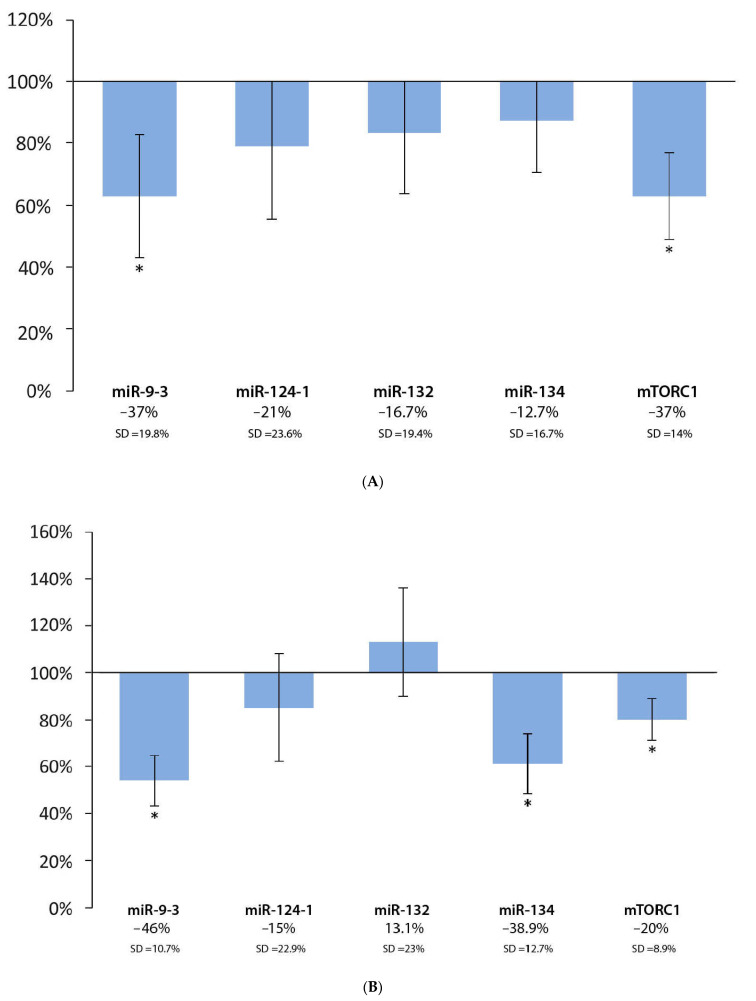
Expression patterns of miR-9-3, miR-124-1, miR-132, miR-134, and mTORC1 in the liver (**A**), spleen (**B**), and kidneys (**C**) of mice treated with DMBA and coffee extract (*n* = 6), compared to the DMBA-induced (*n* = 6) expression (* *p* < 0.05).

**Figure 5 cells-11-01300-f005:**
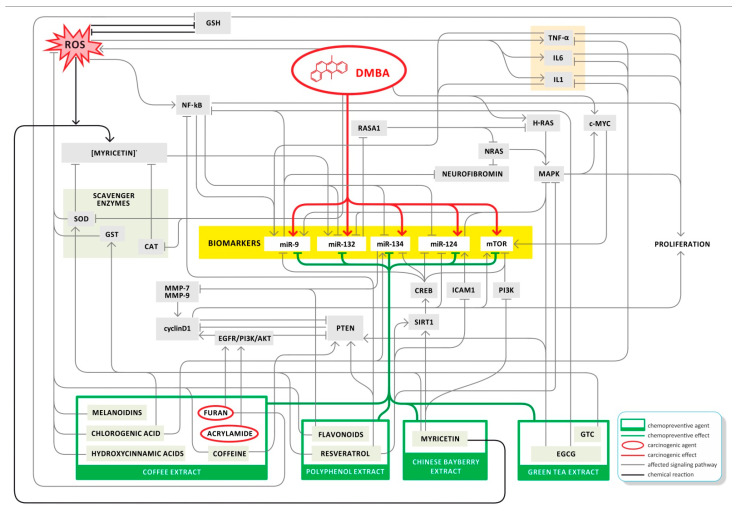
Summary of the relevant factors influencing *miRs* and *mTOR* expression.

**Table 1 cells-11-01300-t001:** The details of the experimental arrangement and applied compounds.

Group	Ip. DMBA	Daily Dose (of 1 Animal)	Producer	Product and Main Components	Latin/Scientific Names	Quantity
**Negative** **Control** (*n* = 6)	-					
**DMBA** **Control** (*n* = 6)	+					
**Flavonoid** **Extract** **+ DMBA** (*n* = 6)	**+**	**30 mg**	**Slimbios Ltd.**	**FruitCafe^TM^**		
				Common grape vine seed, peel	*Vitis vinifera* ‘*Cabernet Sauvignon*’	20 g/100 mL
				Erithritol	(2R,3S)-Butane-1,2,3,4-tetrol	12 g/100 mL
				Resveratrol	*trans*-3,5,4′-Trihydroxystilbenetrans-3,5,4′-trihydroxystilbene	4 g/100 mL
				Blackberry ‘thornfree’ seed, peel	*Rubus fruticosus* “*thorn-free*”	2 g/100 mL
				Blackcurrant seed, peel	*Ribes nigrum*	2 g/100 mL
				Total polyphenol		4000–5000 mg/100 mL
**Green Tea** **Extract** **+ DMBA** (*n* = 6)	**+**	**4 mg**	**Xi’an Longze** **Biotechnology** **Co. Ltd.**	**Green tea**	*Camellia sinensis*	
				Total polyphenol		98.53%
				Total catechins		80.42%
				EGCG	Epigallocatechin-3-gallate	50.45%
				Caffeine	1,3,7-Trimethylxanthine	0.28%
**Coffee****Extract** **+ DMBA** (*n* = 6)	**+**	**30 mg**	**Xi’an Longze** **Biotechnology** **Co. Ltd.**		*Coffee arabica*	
				Chlorogenic acid	3-Caffeoylquinic acid	5.03%
				Caffeine	1,3,7-Trimethylxanthine	1.21%
**Chinese Bayberry** **Extract** **+ DMBA** (*n* = 6)	**+**	**2.5 mg**	**Xi’an Longze** **Biotechnology** **Co. Ltd.**	**Chinese bayberry**	*Myrica rubra*	
				Myricetin	3,5,7,3′,4′,5′-Hexahydroxyflavone	80.42%

**Table 2 cells-11-01300-t002:** Displays of the mTORC1 gene primer sequences (5′-3′), as well as miR-134, miR-132, miR-124-1, miR-9-3, and the internal control (mouse U6 gene).

miR	Forward	Reverse
miR-134	TGTGACTGGTTGACCAGAGG	GTGACTAGGTGGCCCACAG
miR-132	ACCGTGGCTTTCGATTGTTA	CGACCATGGCTGTAGACTGTT
miR-124-1	TCTCTCTCCGTGTTCACAGC	ACCGCGTGCCTTAATTGTAT
miR-9-3	GCCCGTTTCTCTCTTTGGTT	TCTAGCTTTATGACGGCTCTGTGG
mTORC1	AAGGCCTGATGGGATTTGG	TGTCAAGTACACGGGGCAAG
mouse U6	CGCTTCGGCAGCACATATAC	TTCACGAATTTGCGTGTCAT

**Table 3 cells-11-01300-t003:** Summary table of expression changes caused by feeding in the observed DMBA pretreated organs (***** decreasing significantly *p* < 0.05; ******* decreasing significantly *p* < 0.001; **D** decrease was not significant; **O** decrease was questionably; **I** increase was questionably).

	miR-9-3	miR-124-1	miR-132	miR-134	mTORC1
	**Polyphenol extract**				
**Liver**	*	***	***	***	***
**Spleen**	*	*	***	***	***
**Kidneys**	***	*	D	***	***
	**Green tea**				
**Liver**	*	***	*	***	***
**Spleen**	***	***	***	*	***
**Kidneys**	*	*	***	***	***
	**Chinese bayberry**				
**Liver**	***	*	D	*	***
**Spleen**	*	*	*	***	***
**Kidneys**	*	*	***	*	*
	**Coffee extract**				
**Liver**	*	O	O	O	*
**Spleen**	*	O	I	*	*
**Kidneys**	*	*	I	*	*

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
