# Peer review of "The Chemopreventive Effects of Polyphenols and Coffee, Based upon a DMBA Mouse Model with microRNA and mTOR Gene Expression Biomarkers"

_cells, 2022, doi:10.3390/cells11081300_

Round 1
Reviewer 1 Report
mice. Despite interesting data, it has some concerns as follows:
- This study showed RT-PCR data targeting miR-134, miR-132, miR-124-1, miR-9-3 and mTOR in several organs of DMBA treated mice without any mechanism study. It’s a screening data
- It is difficult for readers to find out the novelty compared to previous published papers.
- There are many English flaws including English expression, spelling.
Author Response
Dear Reviewer,
The authors would like to thank for the effort to read and review our manuscript.
Point 1: This study showed RT-PCR data targeting miR-134, miR-132, miR-124-1, miR-9-3 and mTOR in several organs of DMBA treated mice without any mechanism study. It’s a screening data
Response 1: Only a few studies have dealt with the effect of DMBA on the expression of microRNAs (among others, our current and previous research group significantly contributed to this area (1,2,3,4)). Some studies measured the microRNA expressions as parts of a regulatory pathway (5), but others used miRNA expression changes as cellular biomarkers indicating the shift from normal state towards malignant transformation (6) (or miRNA expressions were measured in the induced tumor (7)). A few studies investigated the modulating effect of certain compounds of the mentioned DMBA-miRNA expression patterns (8-10). The role of several microRNAs in cellular regulatory mechanisms and/or signal transmission has been established, this is why we did not consider it essential to measure the expression of all the miRNA target genes or expression of genes involved in a particular miRNA-related pathway. While the previous studies mainly focused on DMBA induced oral cancer where the carcinogenesis could be initiated and promoted by topical application of DMBA onto the oral mucosa, our studies used a systemic gavage (i.e. the intraperitoneal injection entered the circulation rapidly and thus the DMBA exposed different distant organs), and led to the change of expression of different miRNAs there. In our opinion this seems to be relevant new information, because it confirms the possibility of application of these miRNA expressions as biomarkers of systemic carcinogenic exposures. MiRNAs seem to be excellent biomarkers, since they may interfere to more than one regulatory pathways, and thus they may serve as a kind of integrated biomarkers.
In our opinion the major novelty of our study is that we applied the miRNA expressions (which were previously confirmed to be affected by DMBA treatment) as biomarkers of the chemopreventive effects of different phytochemicals. This use of microRNAs is in a very early, experimental phase, as the previously cited papers indicate (8-10). A rapid and effective test system which could appropriately indicate the possible chemopreventive effect of phytochemicals (and of other substances, effects) on miRNA level would make their identification and characterization much simpler and more efficient than the currently used methods. Besides these mentioned advancements our studies identified new miRNA-DMBA-phytochemical relationships as well.
Point 2: It is difficult for readers to find out the novelty compared to previous published papers.
Response 2: We are grateful for your valuable suggestions and added a brief definition and description about the mechanism how miRNAs function and also gave more information about the effect of DMBA.
Point 3:There are many English flaws including English expression, spelling.
Response 3: We highlighted the novel results of our study for better understanding.
Our paper was modified as well as the English language and style was edited about all the valuable comments. Thank you again for your review, and we hope we were able to addresse all the required issues.
- Tomesz et al: Changes in miR-124-1, miR-212, miR-132, miR-134, and miR-155 Expression Patterns after 7,12-Dimethylbenz(a)anthracene Treatment in CBA/Ca Mice. Cells. 2022 Mar 17;11(6):1020.
- Tomesz et al: Effect of 7,12-Dimethylbenz(α)anthracene on the Expression of miR-330, miR-29a, miR-9-1, miR-9-3 and the mTORC1 Gene in CBA/Ca Mice. In Vivo. Sep-Oct 2020;34(5):2337-2343.
- Juhasz et al: Very early effect of DMBA and MNU on microRNA expression. In Vivo. Jan-Feb 2013;27(1):113-7.
- Juhasz et al: In Vivo. Jan-Feb 2012;26(1):113-7. DMBA induces deregulation of miRNA expression of let-7, miR-21 and miR-146a in CBA/CA mice
- Ma et al: Loss of the miR-21 allele elevates the expression of its target genes and reduces tumorigenesis Proc Natl Acad Sci U S A. 2011 Jun 21;108(25):10144-9.
- Devlin et al: Stage-Specific MicroRNAs and Their Role in the Anticancer Effects of Calorie Restriction in a Rat Model of ER-Positive Luminal Breast Cancer. PLoS One. 2016 Jul 19;11(7):e0159686.
- Yu et al: The expression profile of microRNAs in a model of 7,12-dimethyl-benz[a]anthrance-induced oral carcinogenesis in Syrian hamster. J Exp Clin Cancer Res. 2009 May 13;28(1):64.
- Tiwari et al: Preventive effects of butyric acid, nicotinamide, calcium glucarate alone or in combination during the 7, 12-dimethylbenz (a) anthracene induced mouse skin tumorigenesis via modulation of K-Ras-PI3K-AKTpathway and associated micro RNAs. Biochimie. 2016 Feb;121:112-22.
- Baba et al: Blueberry inhibits invasion and angiogenesis in 7,12-dimethylbenz[a]anthracene (DMBA)-induced oral squamous cell carcinogenesis in hamsters via suppression of TGF-β and NF-κB signaling pathways. J Nutr Biochem. 2016 Sep;35:37-47.
- Wahyuniari et al: The Effect of (E)-1-(4’-aminophenyl)-3-phenylprop-2-en-1-one on MicroRNA-18a, Dicer1, and MMP-9 Expressions against DMBA-Induced Breast Cancer. Asian Pac J Cancer Prev. 2020 May; 21(5): 1213–1219.
Reviewer 2 Report
General comments:
In this study the preventive effects of several polyphenols from several extracts were examined in a DMBA treated mice model, to elucidate the effects of chemopreventive agents on the expression profile of the mentioned miRs.
Major comments:
The authors only performed several miRNA expressions for mice treated with or without DMBA. This figure 5 is not validated completely connecting to different extracts. Other non-miRNA expressions were not examined in this study. They links the miRNA and extracts by literature finding. For example, PTEN, AKT, SIRT1, CREB… and others were not found in this study.
Minor comments:
- [Abstract] The species names should be typed in italic fonts.
- Figure legend for Figure 1-3: Please add the repeat number and data meaning such as data = mean+_ SD (n =3).
Author Response
Dear Reviewer,
The authors would like to thank for the thorough review of our paper and for all the valuable comments.
Major comments:
The authors only performed several miRNA expressions for mice treated with or without DMBA. This figure 5 is not validated completely connecting to different extracts. Other non-miRNA expressions were not examined in this study. They links the miRNA and extracts by literature finding. For example, PTEN, AKT, SIRT1, CREB… and others were not found in this study.
Response to major comment:
Only a few studies have dealt with the effect of DMBA on the expression of microRNAs (among others, our current and previous research group significantly contributed to this area (1,2,3,4)). Some studies measured the microRNA expressions as parts of a regulatory pathway (5), but others used miRNA expression changes as cellular biomarkers indicating the shift from normal state towards malignant transformation (6) (or miRNA expressions were measured in the induced tumor (7)). A few studies investigated the modulating effect of certain compounds of the mentioned DMBA-miRNA expression patterns (8-10). The role of several microRNAs in cellular regulatory mechanisms and/or signal transmission has been established, this is why we did not consider it essential to measure the expression of all the miRNA target genes or expression of genes involved in a particular miRNA-related pathway. While the previous studies mainly focused on DMBA induced oral cancer where the carcinogenesis could be initiated and promoted by topical application of DMBA onto the oral mucosa, our studies used a systemic gavage (i.e. the intraperitoneal injection entered the circulation rapidly and thus the DMBA exposed different distant organs), and led to the change of expression of different miRNAs there. In our opinion this seems to be relevant new information, because it confirms the possibility of application of these miRNA expressions as biomarkers of systemic carcinogenic exposures. MiRNAs seem to be excellent biomarkers, since they may interfere to more than one regulatory pathways, and thus they may serve as a kind of integrated biomarkers.
In our opinion the major novelty of our study is that we applied the miRNA expressions (which were previously confirmed to be affected by DMBA treatment) as biomarkers of the chemopreventive effects of different phytochemicals. This use of microRNAs is in a very early, experimental phase, as the previously cited papers indicate (8-10). A rapid and effective test system which could appropriately indicate the possible chemopreventive effect of phytochemicals (and of other substances, effects) on miRNA level would make their identification and characterization much simpler and more efficient than the currently used methods. Besides these mentioned advancements our studies identified new miRNA-DMBA-phytochemical relationships as well.
We are grateful for your valuable suggestions and added a brief definition and description about the mechanism how miRNAs function and also gave more information about the effect of DMBA and added the appropriate literature citation.
- Tomesz et al: Changes in miR-124-1, miR-212, miR-132, miR-134, and miR-155 Expression Patterns after 7,12-Dimethylbenz(a)anthracene Treatment in CBA/Ca Mice. Cells. 2022 Mar 17;11(6):1020.
- Tomesz et al: Effect of 7,12-Dimethylbenz(α)anthracene on the Expression of miR-330, miR-29a, miR-9-1, miR-9-3 and the mTORC1 Gene in CBA/Ca Mice. In Vivo. Sep-Oct 2020;34(5):2337-2343.
- Juhasz et al: Very early effect of DMBA and MNU on microRNA expression. In Vivo. Jan-Feb 2013;27(1):113-7.
- Juhasz et al: In Vivo. Jan-Feb 2012;26(1):113-7. DMBA induces deregulation of miRNA expression of let-7, miR-21 and miR-146a in CBA/CA mice
- Ma et al: Loss of the miR-21 allele elevates the expression of its target genes and reduces tumorigenesis Proc Natl Acad Sci U S A. 2011 Jun 21;108(25):10144-9.
- Devlin et al: Stage-Specific MicroRNAs and Their Role in the Anticancer Effects of Calorie Restriction in a Rat Model of ER-Positive Luminal Breast Cancer. PLoS One. 2016 Jul 19;11(7):e0159686.
- Yu et al: The expression profile of microRNAs in a model of 7,12-dimethyl-benz[a]anthrance-induced oral carcinogenesis in Syrian hamster. J Exp Clin Cancer Res. 2009 May 13;28(1):64.
- Tiwari et al: Preventive effects of butyric acid, nicotinamide, calcium glucarate alone or in combination during the 7, 12-dimethylbenz (a) anthracene induced mouse skin tumorigenesis via modulation of K-Ras-PI3K-AKTpathway and associated micro RNAs. Biochimie. 2016 Feb;121:112-22.
- Baba et al: Blueberry inhibits invasion and angiogenesis in 7,12-dimethylbenz[a]anthracene (DMBA)-induced oral squamous cell carcinogenesis in hamsters via suppression of TGF-β and NF-κB signaling pathways. J Nutr Biochem. 2016 Sep;35:37-47.
- Wahyuniari et al: The Effect of (E)-1-(4’-aminophenyl)-3-phenylprop-2-en-1-one on MicroRNA-18a, Dicer1, and MMP-9 Expressions against DMBA-Induced Breast Cancer. Asian Pac J Cancer Prev. 2020 May; 21(5): 1213–1219.
Indeed, some information (about PTEN, AKT, etc.) mentioned in figure 5. appear only later in the text and we are grateful for that you pointed out this flaw. For better understanding and to avoid confusing feature we moved the figure 5 to the end of discussion, after all the detailed information are mentioned in the text.
Minor comments:
- [Abstract] The species names should be typed in italic fonts.
- Figure legend for Figure 1-3: Please add the repeat number and data meaning such as data = mean+_ SD (n =3).
Response to minor comments:
- Thank you very much for your suggestion. We corrected those mistakes.
- We added the required information about into the text where it was appropriate for better understanding and on the figures we indicated the SD values.
Thank you again for your valuable comments, and we hope we were able to addresse all the required issues.
Reviewer 3 Report
The manuscript is well structured and is based on research from the scientific literature. The authors highlighted the aims, significance and novelty of their study. The paper studies the carcinogen effect of DMBA, as well as the anticarcinogen effects of the examined polyphenol extract, green tea extract, Chinese bayberry extract and coffee extract. The results of the study contribut to the research of chemoprevention by assuming that regular consumption of a diet abundant in polyphenols, as well as coffee exerts anti-inflammatory and anticancer effects.
I suggest to the authors that in figures 1, 2 and 3, the scale of the “y” axis be out of 20 into 20, and in all 4 figures, the background of the “x” and “y” axes be thickened, for a clearer visibility.
Also, in the legend in figure 4, “*** p <0.001”, it must be deleted, because it does not appear on the graph:
Figure 4. Expression patterns of miR-9-3, miR-124-1, miR-132, miR-134 and mTORC1 in the liver 227 (A), spleen (B) and kidneys (C) of mice treated with DMBA and coffee extract compared to the 228 DMBA-induced expression (* p < 0.05; *** p < 0.001)
In general, the quality of the article is good and, overall, the manuscript is interesting to readers. English language and style are good, but there are some minor spelling mistakes. In conclusion, I consider the article could be a useful contribution to the journal. I recommend the manuscript for being published.
Author Response
Dear Reviewer,
The authors would like to thank for all the valuable comments and for the thorough review of our paper.
We carried out your valuable suggestions to increase visibility.
We also deleted the unnecessary information in the legend in figure 4.
Furthermore, we improved the text, corrected the spelling failures.
Thank you again for your thorough review, and we hope we were able to addresse all the required issues.
Round 2
Reviewer 1 Report
MUCH IMPROVED
Reviewer 2 Report
All reviewer's concerns have been well responded.